# Experimental and Analytical Study on Residual Stiffness/Strength of CFRP Tendons under Cyclic Loading

**DOI:** 10.3390/ma13245653

**Published:** 2020-12-11

**Authors:** Chao Wang, Jiwen Zhang

**Affiliations:** 1School of Civil Engineering, Southeast University, Nanjing 210096, China; chao.wang@seu.edu.cn; 2Key Laboratory of Concrete and Prestressed Concrete Structures of the Ministry of Education, Southeast University, Nanjing 210096, China

**Keywords:** Carbon Fiber Reinforced Polymer (CFRP) tendon, fatigue damage model, residual stiffness, residual strength, three-stage degradation

## Abstract

Based on tension–tension fatigue tests, this paper investigated the mechanical property degradation of carbon fiber reinforced polymer (CFRP) tendons from a macroscopic perspective. According to the degradation regularity, this paper proposed a normalized phenomenological fatigue model based on the residual stiffness/strength of CFRP tendons during the fatigue loading process. In this paper, the residual stiffness of CFRP tendons were tested at five stress ranges, while the residual strength was tested at four stress ranges. In order to validate the reliability and applicability of proposed fatigue damage model, the predictions of proposed model and cited models from the literature are discussed and compared. Furthermore, experimental results from literatures were adopted to verify the accuracy of the proposed model. The results showed that the proposed model is applicable to predict both residual stiffness and residual strength throughout fatigue life cycle and has a better accuracy than models from the literature. Moreover, the three-stage degradation can be observed from the degradation processes of stiffness and strength at each stress level.

## 1. Introduction

Fiber reinforced polymer (FRP) composite is a high-performance material achieved by mixing fibers with resin matrix through a proprietary re-engineering process. Due to its several unique characteristics, such as lightweight, high strength, high corrosion resistance, etc., the vigorous development of FRP composites has been promoted. FRP composites have been vastly utilized in various bridge structures, such as external prestressed tendons in Hisho Bridge [1], internal prestressed tendons in Bridge Street Bridge [2], cable-stayed bridge using CFRP cables [3], and the rapid development and application of composite materials started in the 1940s. Glass fiber reinforced polymer (GFRP) composites were first applied, and subsequent technology development and market demand promoted the utilization of higher performance reinforcing fiber. Carbon fiber took the lead in the 1960s due to its superior processing capabilities and lower cost. CFRP has become a notable material applied in engineering structures. It can be used either for retrofitting to strengthen existing structures or as the alternative reinforcing (or pre-stressing) material instead of steel.

Fatigue research on the FRP composite increased in the 1970s. Due to the anisotropy, brittleness, and heterogeneity of composite materials, the failure mechanism and failure characteristics of the FRP composite subjected to fatigue loading are significantly different from those of metallic materials. The fatigue damage of the FRP composite is manifested as the initiation and development of micro-cracks, the degradation of mechanical properties, etc. Degrieck and Van Paepegem [4] classified fatigue damage models of the FRP composite into three major categories: fatigue life models which use S-N curves or Goodman-type diagrams; phenomenological models for residual stiffness/strength; and progressive damage models. From a microscopic perspective, the damage development process of FRP composites under fatigue loading is complicated. Owing to that the interaction of various damage modes is difficult to quantify. Therefore, this paper performs the fatigue analysis of CFRP composites based on macroscopic phenomenological models, i.e., the residual stiffness model and residual strength model. 

Residual stiffness is a macroscopic parameter to characterize the fatigue performance of the FRP composites. Whitworth [5] conducted experimental tests of graphite/epoxy laminates in order to investigate the effects of fatigue loading on the stiffness and strength degradation, assumed that the degradation rate of residual stiffness/strength was related with stress range, and proposed a model to relate the residual stiffness and strength to the loading cycles at a given stress level. Yang et al. [6,7] assumed that the degradation rate of residual stiffness is in a power-law relationship with loading cycles, and proposed a residual stiffness model to predict the statistical distribution of the residual stiffness of composite laminates subjected to cyclic loading, and presented the linear regression analysis method and the Bayesian approach to predict the stiffness degradation of graphite/epoxy laminates. Based on the comparisons of fatigue life and stiffness reduction between glass/phenolic and glass/polyester, Echtermeyer et al. [8] proposed a residual stiffness model in a logarithm form. Philippidis and Vassilopoulos [9] studied the fatigue behavior of glass/polyester laminates under various fatigue loadings, and presented an empirical residual stiffness model based on experimental results. Yao et al. [10,11] proposed a two-parameter residual stiffness model to characterize the stiffness degradation of composite laminates under cyclic loading, and presented a new fatigue life prediction method which can only use tested residual stiffness account for 20% of the fatigue life. Ye [12], Liu et al. [13], and Zhang et al. [14] assumed that the damage rate is in a power-law relationship with maximum stress. Moreover, multiple forms of residual stiffness models are proposed based on investigations of FRP composites subjected to fatigue loading [15,16,17,18,19,20,21]. 

Residual strength is another macroscopic parameter to characterize the fatigue performance of the FRP composites. Broutman and Sahu [22] assumed that the residual strength degrades linearly with the normalized loading cycle based on the test of GFRP laminates. However, experimental results showed that the strength degradation of other FRP composites is nonlinear. Yao and Himmel [23] thought that when FRP composites are subjected to tensile fatigue loading or compressive fatigue loading, the strength degradation has a different nonlinear regularity. Schaff and Davidson [24] proposed a residual strength model, in order to predict the residual strength and fatigue life of FRP composites under constant amplitude or two-stress level cyclic loading. This model is expressed as the form of a power function and displayed that the residual strength monotonically decreases with the loading cycle. For predicting the strength degradation of composites subjected to fatigue loading, Stojković et al. [25] recommended a simplified two-parameter residual strength model, which is verified through adopting different experimental datasets from the literature. Yang and Liu [26,27] assumed that the strength degradation rate is related with the loading cycle and stress level, and proposed a residual strength model to depict this relationship. Subsequently, this model was extended by Hashin [28]. Moreover, Halpin et al. [29], Owen and Howe [30], Revuelta et al. [31], Cremona [32] Zhao et al. [33], Möller et al. [34], and D’Amore et al. [35] investigated the reliability of several residual strength models. Most of the residual strength models proposed in the literature showed that maximum stress is the critical parameter.

The above mentioned residual stiffness models and residual strength models demonstrated the property degradation regularity under cyclic loading from a macroscopic perspective, and accordingly provided theoretical bases for research works on the fatigue property of FRP composites. However, the fatigue models proposed in above literature cannot be applicable to not only predict residual strength but also predict residual stiffness of FRP composites under cyclic loading. 

This paper investigated the stiffness and strength degradation of CFRP tendons based on tension–tension fatigue tests. It can be observed from the aforementioned literature that the residual stiffness and residual strength of FRP composites have a potential relationship with stress range and maximum stress. Therefore, in order to investigate the influence of stress range on mechanical property degradation, the maximum stress was kept consistent. The residual stiffness of CFRP tendons were measured at five stress ranges, while the residual strength was measured at four stress ranges. Based on the degradation regularity in the experiment, this paper proposed a normalized phenomenological model that can be used to predict both the residual stiffness and residual strength of CFRP tendons subjected to cyclic loading. In order to investigate the reliability of this proposed model, comparisons between the predictions of residual stiffness/strength models reported in the literature were performed and of this model were performed as well. Furthermore, this paper investigated the influence of stress range on stiffness/strength degradation regularity at a given maximum cyclic stress.

## 2. Experiment

### 2.1. Material Properties

CFRP tendons tested in this paper are manufactured by Hengshen Co., Ltd. (Zhenjiang, China), which is pultruded from T700-12K carbon fiber and epoxy resin. The fiber volume ratio of this CFRP tendon is 65%. The CFRP tendon has a smooth round surface with a diameter of 8mm and is held by a wedge-type anchorage system. A servo-hydraulic fatigue testing machine, Instron 8800, is adopted to carry out the experiment. Figure 1 depicts the schematic diagram and photograph of the experiment system. It can be found that the anchorage system is clamped by the mechanical wedge grips in Instron 8800 (Instron, Norwood, MA, USA). Static tensile tests of CFRP tendons were performed according to ASTM D3039/D3039M [36], and adopted the displacement-controlled loading with a loading rate of 1mm/min. The axial strains are measured by strain gauges. Five specimens were tested to obtain static characteristics. Table 1 summarized the mechanical properties of CFRP tendons measured from experiments.

### 2.2. Tension–Tension Fatigue Tests

In the fatigue tests, CFRP tendons were subjected to constant amplitude axial cyclic loadings. Tension–tension fatigue tests of CFRP tendons were conducted following ASTM D3479/D3479M [37], which adopted the force-controlled loading. The loading frequency is an important factor affecting the fatigue performance of FRP composites. However, when the frequency is below 10 Hz, the influence can be neglected [38]. Therefore, in this paper, the frequency of cyclic loading was maintained at 5 Hz, and the waveform was sinusoidal in all cases. Fatigue properties of CFRP tendons were tested under five stress levels, i.e., stress range, ∆σ= 900, 800, 600, 500, and 400 MPa. The maximum stress was kept consistent in all cases ($\sigma_{max}=0.6\sigma_{u,ave}$). Three specimens were tested at each stress level.

Figure 2 depicts the loading procedure of fatigue test, each specimen was first quasi-statically loaded to $0.6\sigma_{u,ave}$ with a loading rate of 1 mm/min. During the quasi-static loading process, longitudinal strains of the CFRP tendon were measured, from which the initial stiffness could be obtained. The initial strength is determined by $\sigma_{u,ave}$. Before cyclic loading, the specimen should be unloaded to $\sigma_{mean}$. In order to characterize degradation, during subsequent cyclic loading process, measurements of residual stiffness and residual strength were performed by a quasi-static process according to ASTM D3039/D3039M [36]. In the tension–tension fatigue tests, axial loading force was directly measured by the built-in force sensor of Instron 8800, while longitudinal strain was measured by both the strain gauge and the extensometer. Strain gauges were destroyed after several loading cycles. Therefore, the extensometer is recommended for strain measurements in the quasi-static process after certain loading cycles.

When the cyclic loading achieves a given cycle, $n_{i}$, the specimen will be unloaded. Then, the quasi-static loading process to measure the initial stiffness will be performed again, for measuring the residual stiffness, $S(n_{i})$. After the quasi-static process, the cyclic loading is carried on until next given cycle, $n_{i+j}$. The above procedure will be repeated until fatigue failure of the CFRP tendon. The fatigue life of this specimen can be obtained from the accumulation of loading cycles at this stress level. 

Based on the above obtained fatigue life, $N_{i}$, the loading cycles for the measurements of residual strength can be determined. When the loading cycle achieves the given fraction of fatigue life, 
$n_{i}/N_{i}$, such as 15%, 20%, 50%, the specimen will be unloaded. Then, this specimen will be quasi-statically loaded until failure with a loading rate of 2 mm/min, in order to measure the residual strength, $S\left(n_{i}\right)$ of CFRP tendon. A new specimen will be cyclically loaded to another cycle, $n_{i+j}$. The same quasi-static process will be applied in another specimen to measure the residual strength, $S\left(n_{i+j}\right)$ at another fraction of fatigue life, $n_{i+j}/N_{i}$.

## 3. Fatigue Damage Model

There are various damage mechanisms in the FRP composite, such as matrix cracking, fiber breakage, matrix/fiber debonding, and delamination. During the process of cyclic loading, single or multiple damage mechanisms appear in the FRP composite. The mechanical properties degrade during this process, macroscopically shown as stiffness degradation and strength degradation. Because the damage mechanism of the FRP composite is complicated during the fatigue loading process, it is impractical to establish a fatigue damage model from a microscopic perspective. Based on experimental data, Reifsnider [39] pointed out that fatigue damage evolution of the FRP composite is nonlinear and has a three-stage development regularity, as displayed in Figure 3. In the initial stage of fatigue damage, transverse cracks generate in the matrix. With the accumulation of loading cycles, FRP composite reaches the characteristic damage state (CDS) when transverse matrix cracks become saturated. Stiffness and strength degrade rapidly in this stage. In the second stage, the occurrence of matrix/fiber cracking and delamination leads to stress redistribution, which results in the cyclic loading being mainly carried by the fiber. In the final stage, fatigue damage develops rapidly, and large amounts of brittle fiber breakages cause the sudden failure of the FRP composite.

This paper proposed a normalized phenomenological fatigue model based on the experimental residual stiffness/strength of CFRP tendons during the fatigue loading process. In order to illustrate the reliability and applicability of proposed fatigue model, the predicted results of the proposed model are compared with the predicted results of reported residual stiffness models and residual strength models from the literature, respectively. As previously mentioned, several recommended residual stiffness and residual strength fatigue model are listed below.

### 3.1. Residual Stiffness Model

Whitworth [5] assumed that the rate of residual modulus reduction is inversely proportional to a certain power of the residual modulus itself. Yang [6,7] assumed that the residual stiffness degradation rate is assumed to be power function of the number of loading cycles. Echtermeyer [8] assumed that the modulus reduction can be characterized by the logarithm of the cycle number. Philippidis and Passipoularidis [9] assumed that the residual stiffness is linear with the cycle number. Yao [10,11] proposed a new model in the form of power function. The normalized form of residual stiffness models reported in the literatures in this paper are listed as follows:H. A. Whitworth (1987) [5]:
(1)$$\frac{S\left(n\right)}{S\left(0\right)}=e^{\frac{\ln\left[1-\frac{\beta n}{N}\right]}{\alpha }}$$J. N. Yang et al. (1990) [6]:
(2)$$\frac{S\left(n\right)}{S\left(0\right)}=-\alpha \cdot {\left(n/N\right)}^{\beta }+\gamma$$A. T. Echtermeyer et al. (1995) [8]:(3)$$\frac{S\left(n\right)}{S\left(0\right)}=\alpha -\beta \cdot \log n$$T. P. Philippidis and V. A. Passipoularidis (2000) [9]:(4)$$\frac{S\left(n\right)}{S\left(0\right)}=1-\alpha \cdot \frac{n}{N}$$W. X. Yao et al. (2012) [11]:(5)$$\frac{S\left(n\right)}{S\left(0\right)}=1-\left(1-\frac{S\left(cr\right)}{S\left(0\right)}\right)\left(1-\frac{1-{\left(n/N\right)}^{\alpha }}{{\left(1-n/N\right)}^{\beta }}\right)$$


### 3.2. Redisual Strength Model

Brountman and Sahu [22] assumed that the residual strength degrades linearly with the normalized loading cycle. Scaff and Davidson [24] assumed that the residual strength is in a power function of the loading cycles. Yao and Himmel [23] combined the sine function and cosine function of loading cycles to describe the degradation of residual strength. Philippidis and Passipoularidis [40] proposed a new residual strength model based on probabilistic and deterministic theory, which combines the power function and exponential function of loading cycles. Stojković [25] assumed the residual strength have a power-law relationship with loading cycles. The normalized residual strength models reported in the literatures are listed as follows:L. J. Brountman and S. Sahu (1972) [22]:(6)$$\frac{S\left(n\right)}{S\left(0\right)}=1-\left(1-\frac{S_{max}}{S\left(0\right)}\right)\frac{n}{N}$$J. R. Scaff and B. D. Davidson (1997) [24]:(7)$$\frac{S\left(n\right)}{S\left(0\right)}=1-\left(1-\frac{S_{max}}{S\left(0\right)}\right){\left(n/N\right)}^{\alpha }$$W. X. Yao and N. Himmel (2000) [23]:(8)$$\frac{S\left(n\right)}{S\left(0\right)}=1-\left(1-\frac{S_{max}}{S\left(0\right)}\right)\left[\frac{\sin(\beta n/N)\cos\left(\beta -\alpha \right)}{\sin\beta \cos(\beta n/N-\alpha )}\right]$$T. P. Philippidis and V. A. Passipoularidis (2007) [40]:(9)$$\frac{S\left(n\right)}{S\left(0\right)}=1-\left(1-\frac{S_{max}}{S\left(0\right)}\right){\left(n/N\right)}^{\mathrm{\alpha exp}\left(\beta n/N\right)}$$N. Stojković et al. (2017) [25]:(10)$$\frac{S\left(n\right)}{S\left(0\right)}=1-\left(1-\frac{S_{max}}{S\left(0\right)}\right)[1-{\left(1-{\left(\frac{n}{N}\right)}^{\alpha }\right)}^{\beta }]$$
where $S_{\left(n\right)}$ is residual stiffness/strength of the CFRP tendon after loading cycles; S(0) is the initial stiffness/strength of the CFRP tendon (it is assumed that the initial strength equals to $\sigma_{u,ave}$); S(cr) is the critical stiffness; $S_{max}$ is the maximum stress of the cyclic loading; n is the loading cycle; N is the fatigue life of the CFRP tendon; n/N is fractional life (normalized number of cycles with respect to fatigue life); α, β, γ are model parameters, which are determined by the fitting result.

## 4. Discussion of Experimental Results

The experimental results from fatigue tests were analyzed in this paper and were compared with predicted results of above proposed residual stiffness models from the literature. Figure 4 plotted the stiffness degradation curve of normalized residual stiffness with respect to initial stiffness, S(n)/S(0) versus the normalized number of cycles with respect to fatigue life, n/N. In order to investigate the effect of stress range on stiffness degradation of the CFRP tendon, degradation curves of CFRP tendons under cyclic loading at five different stress ranges (i.e., ∆σ = 900, 800, 600, 500, 400 MPa) were depicted in Figure 4. In addition, the predicted results of models proposed in the literature are plotted.

The above mentioned residual stiffness models from the literature were adopted to fit experimental results, which use the least squares method. The fitting curves of the residual stiffness models from the literature are plotted in Figure 4. A two-stage degradation is presented by Whitworth’s model. Yang’s model is expressed in a form of power function and presents a gradual–rapid two-stage degradation. Echtermeyer’s model is in a form of logarithmic function and presents a rapid–gradual two-stage degradation. Philippidis’s model is in a form of linear function and presents a monotonic linear degradation. Yao’s model can depict the three-stage degradation under several stress ranges. The correlation coefficients, $R^{2}$ of the above mentioned models from the literature are all below 0.7.

Figure 5 plots the strength degradation curve of normalized residual strength with respect to initial strength, S(n)/S(0) versus normalized number of cycles with respect to fatigue life, n/N. In order to investigate the effect of stress range on strength degradation of the CFRP tendon, degradation regularities of CFRP tendons under cyclic loading with four different stress ranges (i.e., ∆σ = 900, 800, 600, 500 MPa) are depicted in Figure 5. In addition, the predicted results of models proposed in the literature are plotted. The correlation coefficients, $R^{2}$ of the aforementioned models are all below 0.7.

## 5. A New Proposed Fatigue Model

It can be observed from Figure 4 and Figure 5 that both residual stiffness and residual strength have the three-stage degradation. However, the fatigue models proposed in the literature cannot predict this regularity. Some researchers recommended that the residual stiffness/strength model have a logarithmic function. Moreover, it can be noticed that f(x)=ln(x) has a monotonically increasing trend in (0, +∞), while g(x)=ln(a−x) has a monotonically decreasing trend in (0, a). The combination of f(x) and g(x) can form a new function which have a three-stage decreasing trend. Based on the degradation regularities of stiffness and strength, this paper proposed a new fatigue model, expressed as:(11)$$\frac{S\left(n\right)}{S\left(0\right)}=\alpha \cdot \ln\frac{n/N}{\gamma -n/N}+\beta$$
where, α, β,  and γ are model parameters determined by stress level and material component.

Table 2 summarizes the parameters of proposed fatigue model by fitting analysis. The predicted results of this new proposed fatigue model were plotted in Figure 4a–e and Figure 5a–d. Based on the predicted degradation curves, Figure 6 plots the curve of corresponding degradation rate. It can be observed that stiffness degrades obviously in the initial stage of cyclic loading. When n/N exceed 15–20%, the residual stiffness of the CFRP tendon tends to be stable; and when  n/N exceed 80%, degradation rates of stiffness increase again.

It can be observed from Figure 4, Figure 5, Figure 6 and Figure 7 both stiffness and strength degrade obviously in the initial stage of cyclic loading. Figure 6 and Figure 7 show that the degradation rate is larger than the rate in second stage. When n/N exceeds 15–20%, the degradation rate of the residual stiffness/strength tends to be stable; and when n/N exceeds 80%, the degradation rates of stiffness/strength increase again, shown as obvious stiffness/strength degradation. Regularities can be obtained from Figure 4, Figure 5, Figure 6 and Figure 7 as follows:(a)Under cyclic loading with various stress ranges, both the stiffness and strength of CFRP tendon have three-stage degradations throughout the fatigue life cycle.(b)Stiffness/strength degradation rate of the CFRP tendon increases with the growth of stress range. This means that if the stress range gets larger, stiffness/strength degradation of the CFRP tendon become more obvious. (c)When the stress range gets smaller, the stiffness/strength degradation rates of initial stage and final stage become closer to that of the second stage.

In order to verify the accuracy of proposed fatigue model, this paper adopted the experimental results from Xu [41] and Post [42]. As shown in Figure 8a, Xu [41] measured the residual stiffness of CFRP laminates during cyclic loading process at three stress levels. Post [42] measured the residual strength of GFRP laminates at three stress levels, as shown in Figure 8b. The new proposed fatigue model in this paper was used to predict the residual stiffness and strength of the composite materials from Xu [41] and Post [42]. It can be observed in Figure 8 that the predictions of both residual stiffness and residual strength have a good agreement with measured results. Therefore, the new proposed fatigue model can reliably predict the mechanical property degradation of FRP composites.

Table 3 summarizes the correlation coefficients of above-mentioned fatigue models and the new proposed model. From the comparisons of correlation coefficients at different stress levels, it can be found that the prediction of this new proposed model has a better agreement with the measured results.

## 6. Conclusions

In order to investigate the mechanical property degradation of composite material subjected to fatigue loading, tension–tension fatigue tests of CFRP tendons were conducted. In this paper, the residual stiffness and residual strength of CFRP tendons were tested at different stress ranges, and a new fatigue damage model was proposed according to the experimental results. The following conclusions can be drawn from the investigations presented here:(a)Both the stiffness and strength of CFRP tendon degrade during the fatigue loading process. Also, it can be observed that as the stress range increases, stiffness and strength of CFRP tendons degrade more obviously.(b)The three-stage regularity can be observed from degradation processes of stiffness and strength when CFRP tendons and other FRP composites subjected to fatigue loading. In the first stage, transverse cracks become more saturated when CFRP tendons subjected to lower stress range. Therefore, in the first stage, mechanical property degrades more obviously at lower range. The damage mechanism in the first stage is matrix cracking and is matrix/fiber debonding in the second stage at lower stress range. However, the damage mechanism in the first and second stage becomes the mixing of matrix cracking, matrix/fiber debonding and fiber breakage at higher stress range. Therefore, the boundaries between adjacent stages become more obvious when stress range decreases.(c)The proposed fatigue damage model is applicable to predict both residual stiffness and residual strength throughout fatigue life cycle. This new proposed model has a better accuracy than the models from the literature based on the experimental results of CFRP tendons and results from the literature.

## Figures and Tables

**Figure 1 materials-13-05653-f001:**
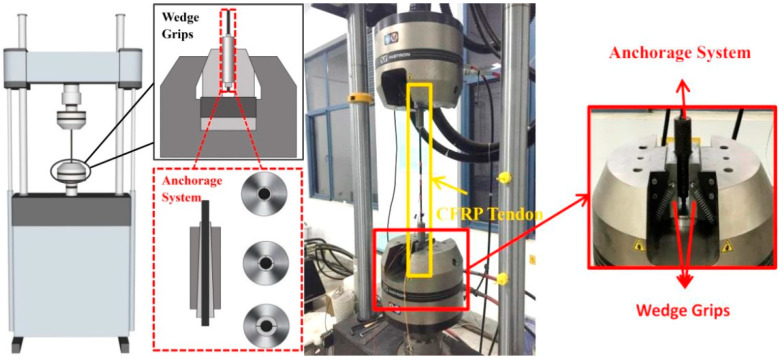
Schematic diagram and photograph of testing system.

**Figure 2 materials-13-05653-f002:**
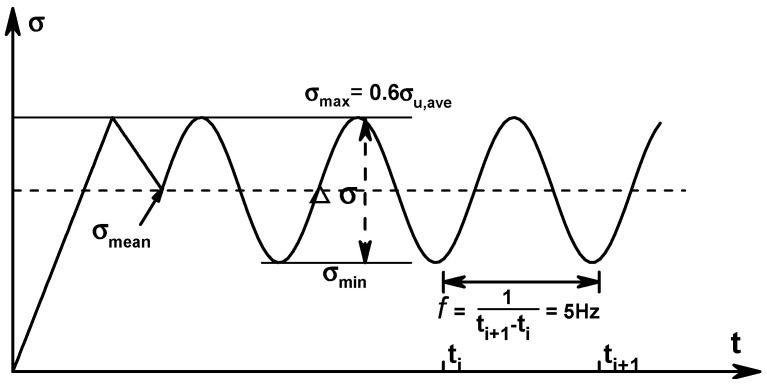
Loading procedure of the fatigue test.

**Figure 3 materials-13-05653-f003:**
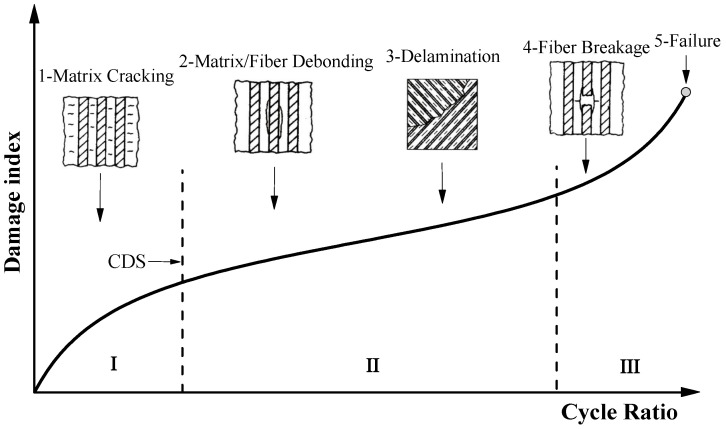
Schematic fatigue damage evolution of the FRP composite.

**Figure 4 materials-13-05653-f004:**
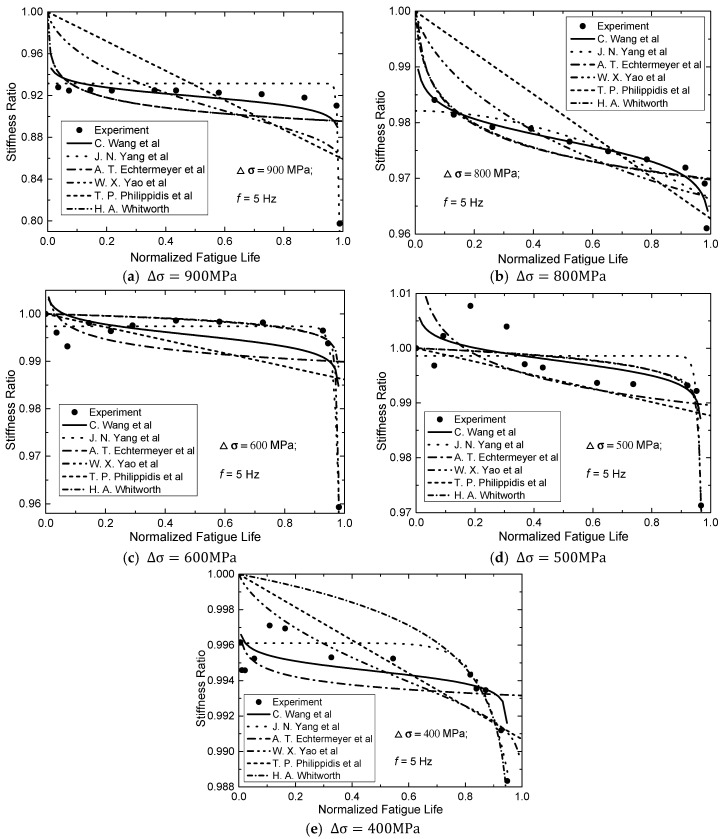
Predicted and measured residual stiffness of CFRP tendons at different stress ranges: (**a**) ∆σ = 900MPa, (**b**) ∆σ = 800MPa, (**c**) ∆σ = 600MPa, (**d**) ∆σ = 500MPa, (**e**) ∆σ = 400MPa.

**Figure 5 materials-13-05653-f005:**
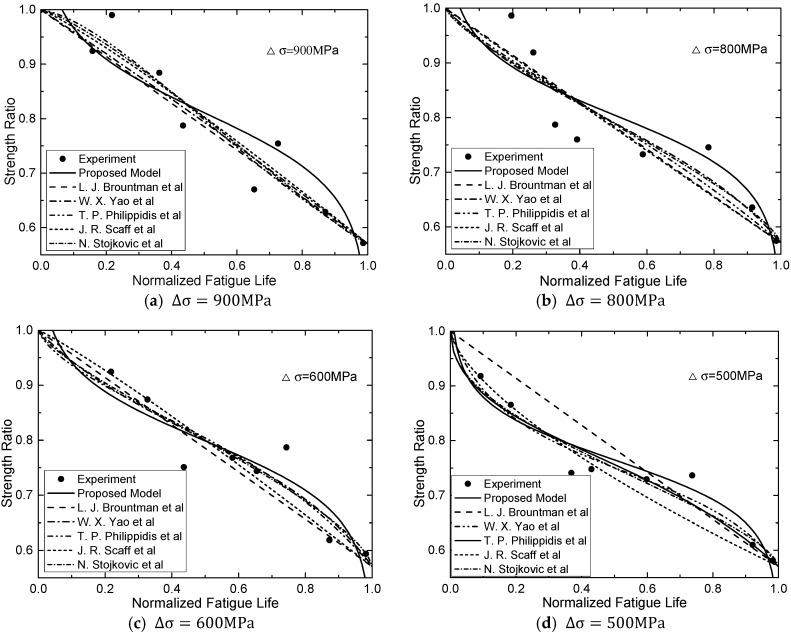
Predictions of residual strength models and measurements of CFRP tendons at different stress ranges: (**a**) ∆σ = 900MPa, (**b**) ∆σ = 800MPa, (**c**) ∆σ = 600MPa, (**d**) ∆σ = 500MPa.

**Figure 6 materials-13-05653-f006:**
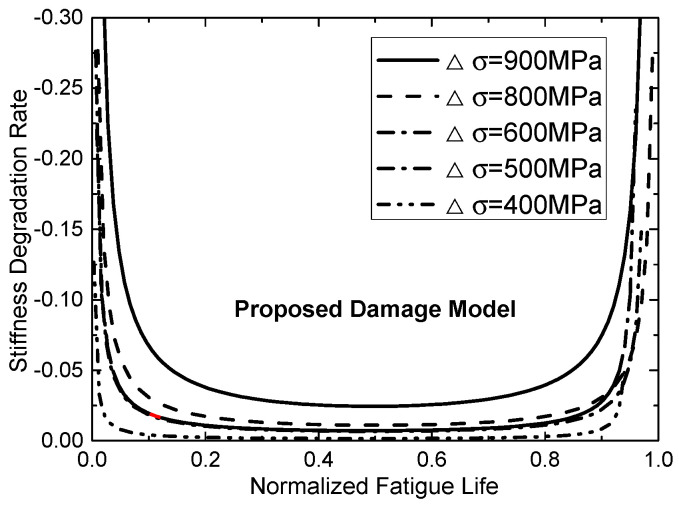
Stiffness degradation rates of CFRP tendons at different stress ranges.

**Figure 7 materials-13-05653-f007:**
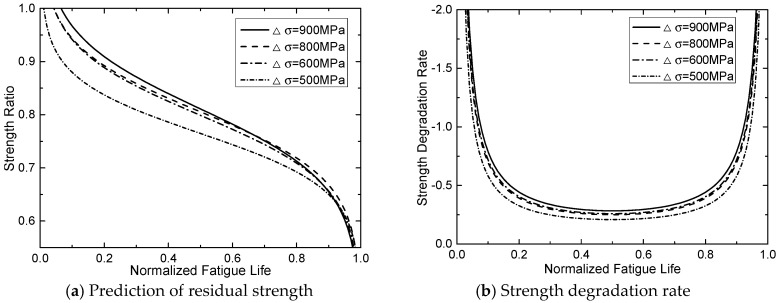
Residual strength predictions of new proposed model: (**a**) prediction of residual strength, (**b**) strength degradation rate.

**Figure 8 materials-13-05653-f008:**
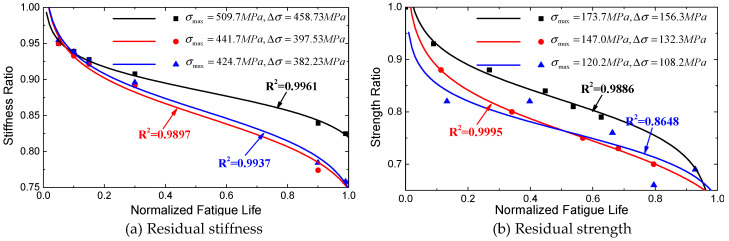
Mechanical property degradations of composite materials under different stress levels: (**a**) residual stiffness, (**b**) residual strength.

**Table 1 materials-13-05653-t001:** Mechanical Properties of CFRP tendons measured from experiments.

Specimen	Young’s Modulus, *E* (GPa)	Ultimate Strength, (MPa)	$E_{ave}$ (Gpa)	$\sigma_{u,ave}$ (Mpa)
S1	155.4	2064.9	155.9	2084.0
S2	162.1	2104.4		
S3	159.5	2142.4		
S4	149.7	2049.2		
S5	152.6	2059.3		

**Table 2 materials-13-05653-t002:** Parameters of the proposed fatigue model.

Stress Range	Residual Stiffness				Residual Strength			
	α	β	γ	$R^{2}$	α	β	γ	$R^{2}$
900	−0.00605	0.98912	0.91945	0.862	−0.07095	0.99923	0.81094	0.823
800	−0.00276	0.98483	0.97684	0.945	−0.06266	0.99941	0.80541	0.776
600	−0.00165	0.98125	0.99557	0.721	−0.06482	0.99968	0.79876	0.768
500	−0.00175	0.96750	0.99769	0.746	−0.05227	0.99971	0.76475	0.794
400	−0.00042	0.94790	0.99454	0.693	N/A			

**Table 3 materials-13-05653-t003:** Correlation coefficients of fatigue models at different stress levels.

Residual Stiffness	Stress Level			Residual Strength	Stress Level		
	Ⅰ	Ⅱ	Ⅲ		Ⅰ	Ⅱ	Ⅲ
Whitworth	0.9674	0.9799	0.9814	Brountman	0.9975	0.9849	0.7920
Yang	0.9917	0.9841	0.9934	Scaff	0.9859	0.9921	0.6712
Echtermeyer	0.9377	0.9223	0.9203	Yao	0.9714	0.9888	0.6005
Philippidis	0.9946	0.9586	0.9675	Philippidis	0.9849	0.9109	0.5772
Yao	0.9958	0.9845	0.9897	Stojković	0.9856	0.9224	0.5820
New model	0.9961	0.9897	0.9937	New model	0.9886	0.9995	0.8648

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
