# Peer review of "Experimental and Analytical Study on Residual Stiffness/Strength of CFRP Tendons under Cyclic Loading"

_materials, 2020, doi:10.3390/ma13245653_

Round 1

Reviewer 1 Report

Fatigue behaviour of CFRP tendons is worthy of investigation. The reviewer regards the reported experimental results as a valuable source for further analysis. However, this submission requires substantial modification before being considered for publication.

The literature review must rationalise the idea of the research. What is the novelty of this study?

The comparative analysis of the models reported in the literature cannot use the same dataset as applied for the development of the proposed model. The authors must quantify the prediction accuracy of the considered models and formulate the means of improving the prediction adequacy.

The discussion of the results must be more "scientific". The authors must describe the mechanisms responsible to the observed fatigue degradation of mechanical properties of CFRP.

The list of references must be updated: the newest two articles published in 2017.

The text contains numerous errors; the authors used the wrong terminology. The attached manuscript defines the required modifications highlighted in the comments.

Reviewer 2 Report

  1. The author presented a new fatigue model based on the degradation regularity. However, the scientific basic and references for formula are not enough. Plz provide more details or rationale for how the new model (formula) was derived.
  2. What is the meaning of the α, β, γ and how is it determined? If they are just fitting parameters for case by case, then how could you predict the fatigue and residual properties?
  3. Is there any specific reason for setting the maximum stress to 0.6 ultimate strength?
  4. For the fatigue test, average stress and stress ratio (R=σmin/σmax) is dominate factor for fatigue life not maximum stress. As the stress range (Δσ) of this paper increases, the average stress decreases and stress ratio is increases because the maximum stress was fixed. Shouldn't the experiment be performed with the average stress rater than the maximum stress as a fixed variable?

Reviewer 3 Report

What a

The topic of the paper is interesting and analyzed materials is very important in our days. But still disadvantages were noticed in the paper:

  1. It is not clear, why just 3 specimens were selected for the static tensile tests? Usually 5 or more specimens are analyzed during experimental tests, in order to have more precise results.
  2. According to which standard specimens were produced? Information about manufacturing processes of the specimens is not provided in the paper. Parameters od the specimens are not presented.
  3. Elements of the Fig. 1 are not presented and explained in the text, even in figure.
  4. Full explanations of the results are missing.
  5. Formatting of the paper is inappropriate.
  6. 90% of the references are older than 10 years.

re the contributions of this paper?

Round 2

Reviewer 1 Report

  1. The manuscript was improved, but the authors did not react to the comments presented in the attached pdf-version of the original article. The authors must check the comments in the text (the pdf-document with comments is attached again to this report) and reflect all critical notes. Please check attachment.
  2. The authors did not eliminate the severe methodology flaw. Reply #2 by the authors ("If we want to improve the accuracy of our proposed model, more fatigue tests should be performed") indicates that. The introduction of the determination coefficients inserted in the manuscript (Lines 229 and 235, and Table 2) does not solve the problem. The same dataset cannot be used for verification of the models presented in the literature and the development of the new model. Such a methodology is wrong; it artificially increases the accuracy of the developed model (concerning the approaches reported in the literature). The accuracy analysis of all models (including the developed model) must use the dataset different that one applied for the development of the proposed model. The authors must find appropriate data in the literature.

Reviewer 2 Report

All review questions were answered well. The rationale for the new fatigue model was also clear.

Author Response

Thanks for your review

Round 3

Reviewer 1 Report

As mentioned in the previous review stage, the fatigue behaviour of CFRP tendons is worthy of investigation. The reported experimental results can be useful for potential readers by the Journal. The reviewer recommends this work for the publication in Materials after minor corrections. The following modifications are required:

  • Line 51. The statement “Residual stiffness is a macroscopic non-destructive parameter…” is misleading. The term “non-destructive” means the absence of mechanical damaging.
  • Lines 190–199, 209–218, and 262. Please add literature references of the models and numbers to the formulas.
  • Line 276. Please delete unnecessary dot.
  • Lines 279–285. The reviewer recommends formatting this text as a numbered or bulleted list. The adjectives “larger” and “smaller” require a reference for the comparison (larger than what does?). Please rephrase these regularities.

Author Response

Dear Reviewer,

Many thanks for your suggestions!

We have made the revisions according to your comments.

Please check them in the revised manuscript.

Sincerely,

Chao Wang & Jiwen Zhang